Morphological variations of the femoral head-neck junction in historical skeletal material

Myszka Anna 1
Kubicka Anna Maria amkkubicka@gmail.com 2 3
1 Institute of Biological Science, Cardinal Stefan Wyszyński University in Warsaw , Warszawa , Poland
2 Department of Zoology, Poznań University of Life Sciences , Poznań , Poland
3 Département Homme et Environnement, Museum national d’Histoire naturelle , Paris , France
Reno Philip
Electronic publication date: 2025 Oct 16
Publication date: 2025
Volume: 13
Electronic Location ID: e20236
Received 2025 May 28; Accepted 2025 Sep 23
Copyright: ©2025 Myszka and Kubicka
Copyright year: 2025
Copyright holder: Myszka and Kubicka
License: This is an open access article distributed under the terms of the Creative Commons Attribution License, which permits unrestricted use, distribution, reproduction and adaptation in any medium and for any purpose provided that it is properly attributed. For attribution, the original author(s), title, publication source (PeerJ) and either DOI or URL of the article must be cited.
License URL: https://creativecommons.org/licenses/by/4.0/

Keywords: Computed-tomography, Poirier’s facet, Allen’s fossa, Femoral plaque, Geometric morphometrics, 3D model, Femur, Acetabulum, Anatomical analysis

Funding: Cardinal Stefan Wyszyński University in Warsaw (Poland) The project DEC-IND-8/23 was founded by Cardinal Stefan Wyszyński University in Warsaw (Poland). The funders had no role in study design, data collection and analysis, decision to publish, or preparation of the manuscript.

==============================
Background

Poirier’s facet, Allen’s fossa and femoral plague are the main morphological variations of the femoral head-neck junction. The study aimed to answer questions about the association between the shape of the proximal end of the femoral bone and acetabulum in bones with head-neck junction changes and the differences in shape and size between joints with the above changes and joints without ones.

Methods

The analyses were performed on the computed tomography scans (CTs) of the 52 sets of bones (femur and pelvic bone) from the Polish skeletal material dated to the 14th–19th centuries. Based on CTs, three-dimensional models of the femurs and pelvic bones were created and then analysed using linear measurements and a geometric morphometric approach. Analysis of variance (ANOVA) was calculated to analyse differences in size; in turn, canonical variate analysis (CVA) was calculated to investigate changes in shape between bones with femoral-neck changes and bones without ones.

Results

According to the CVA, there were no significant differences in shape between bones with Allen’s fossa, femoral plaque, or Porier’s facet and hip joints without any observable changes (p > 0.05). Bones with observable Allen’s fossa, femoral plaque, Porier’s facet and hip joints without changes showed similar variations in femoral head shape. The difference was in the femoral head height between bones femoral plaque and bones with Allen’s fossa (p = 0.047, mean difference = 3.78 mm). Acetabula in the sets of bones without head-neck junction changes showed slightly lower shape variation than acetabulum in the sets of bones with changes. In joints with head-neck junction changes, a more indented antero-posterior part of the lunate surface and indented inferior edge along its entire length were observed.

Conclusions

Geometric morphometrics and measurements showed similarities in the shape of the joints with and without changes in head-neck junction region. This may indicate that morphological changes in the femoral head-neck junction do not significantly affect the morphology of the femur and acetabulum. However, understanding the role and efficiency of this influence needs further studies.

Introduction

Osseous non-metric variations of the femoral head-neck junction have been the subject of anatomical and anthropological studies for many years (Villotte & Knüsel, 2009; Ghosh, Sethi & Vasudeva, 2014; Verna et al., 2014; Göhring, 2021). Among them, certain morphological features of the anterior aspect of the femoral head-neck junction (Poirier’s facet (PO), Allen’s fossa (AL), and femoral plaque (FP)) have been considered (Barnes, 2012; Mellado & Radi, 2015). Disagreement on descriptions, terminology, the lack of standardised scoring methods, and poor knowledge of the variability of the traits make it difficult to interpret their meaning (Radi et al., 2013).

Poirier’s facet is defined as an extension of the articular surface of the femoral head through the anterosuperior part of the femoral neck (Angel, 1964; Villotte & Knüsel, 2009; Verna et al., 2014; Zurmühle et al., 2017). The surface has a smooth contour. Its profile is continuous and runs from the femoral head to the femoral neck region without a visible transition. It can have an oval or triangular shape (Villotte & Knüsel, 2009; Radi et al., 2013). Femoral plaque is an imprint which is located on the anterior margin of the femoral neck (close to the head). The change may be delimited by a distinct border (Radi et al., 2013). Allen’s fossa is located on the anterior part of the femoral neck connected to its junction with the articular surface (El, 1963; Finnegan, 1978; Ji et al., 2014).

The front surface of the proximal end of femoral bone shows great variability in the head-neck junction region morphology, and the three mentioned changes do not show the whole spectrum of possible variations (Mellado & Radi, 2015). The aetiology of these traits has been widely discussed, but no consensus has been reached thus far. As it was mentioned by Radi et al. (2013), the features of the femoral head-neck junction area have long been interpreted as markers of physical activities: flexion and abduction during locomotion (Angel, 1964; Trinkaus, 1975); full thigh extension as in walking/running downhill (Angel, 1964); squatting (Charles, 1893); sitting cross-legged and horseback riding (Molleson & Blondiaux, 1994; Pálfi & Dutour, 1996; Ruff, 1999); lying on one’s side when sleeping, with partial flexion and intra-rotation of one or both thighs (Meyer, 1924; Meyer, 1934). The interpretation of these features is also obscure, with proposals often contradictory to functional hypotheses: pressure on the acetabular rim with the hip in flexion, capsular pull in both flexion and extension or pressure exerted by the m. iliopsoas or by the m. rectus femoris tendon (El, 1963; Angel, 1964; Trinkaus, 1975).

Uncertainty regarding the neck-head junction definitions and aetiology is probably one of the major causes of the variation reported in the literature (Medlej et al., 2021). The prevalence of Poirier’s facet varies between 3% and 71% (El, 1963; Angel, 1964; Vyas et al., 2013; Radi et al., 2013); Allen’s fossa between 5% and 43% (Vyas et al., 2013; Radi et al., 2013); the plaque type between 44% and 90% (Vyas et al., 2013; Radi et al., 2013). Some studies found it to be more prevalent among males (Verna et al., 2014; Bühler & Kirchengast, 2022), while others reported that it is sex-independent (Radi et al., 2013). Disagreement exists regarding their association with age (Radi, 2014) or its laterality (Vyas et al., 2013; Radi, 2014; Göhring, 2021).

Poirier’s facet is connected by some authors to variations in the development of the head of the femur and in the closure of the physeal line. These variations could have chance origin or possibly mechanically driven (Radi, 2014). Due to the morphological variability of the changes, their definitions are inconsistent, which often leads to inappropriate classification of bony changes. It makes it difficult to perform comparative analyses and thus to assess the causes of the presence/absence of intra- and inter-population differences and resolve disputes regarding the aetiology of changes (Kim et al., 2011; Rubin, 2013; Omoumi et al., 2014).

It is known a relationship between Allen’s fossa, Poirier’s facet, femoral plaque and variabilities regarding the shape and size of the acetabulum and proximal femur (Benes et al., 2025). The Poirier’s facet can affect the contact between the head and neck, which may correspond to a larger acetabular protrusion or a larger average head. Allen’s fossa—the configuration and size that can be determined during grinding of the orbicularis zone—which in turn may be related to the neck angle or depth of the acetabulum. A femoral plate, which can often be used with the device, affects the morphology of the femoral head and neck (Barnes, 2012).

Taking the above into consideration, this study was based on the hypothesis that the hip joints without changes and with Poirier’s facet, Allen’s fossa and femoral plague show significant differences in morphology. The research analysed (a) the variation in the shape of the proximal end of the femoral bone and acetabulum in joints with and without analysed changes, (b) differences in size and shape between three types of femoral head-neck junction changes and joints without them. To achieve these purposes, a combination of geometric morphometrics and measurements was used for the above analyses. This comprehensive analysis may enhance the knowledge of the variation in the head-neck junction and contribute to current orthopaedic research on the relationship between the femur and acetabulum.

Material and Methods

The analysed skeletal material comes from a cemetery located in Radom (east-central Poland, about 100 km south of Warsaw, Poland). The settlement of this cemetery dates to the 11th–19th centuries (Trzeciecki, 2018). The skeletal material from Radom is a part of the skeletal collection of the Institute of Biological Sciences Cardinal Stefan Wyszyński University in Warsaw (Poland). According to Polish law, no administrative and ethical permits were required for the described study on historical osteological remains. The authors confirm that all stages of the research (collection, CT scanning and analysis) were performed following the regulations on the analysis of human remains. The osteological collection is stored at the Institute of Biological Sciences, Cardinal Stefan Wyszyński University in Warsaw (Poland), and in turn, 3D models are stored in the Morphosource platform (project ID: 000740996) and are available on request from a corresponding author.

The archaeological research indicates that the Radom region in the historical period was characterized by high settlement potential (Mnich, Lisowska-Gaczorek & Szostek, 2018; Grezak et al., 2018). An important point in Radom’s history that occurred throughout the 14th century was urbanization (urban revolution, social and demographic changes), after the settlement was granted status as a city (Trzeciecki, 2018). Analysis of the diet (using isotopes) and skeletal markers of environmental stress showed that the biological condition of the Radom population was rather good and had not changed significantly since the Late Middle Ages (Mnich, Lisowska-Gaczorek & Szostek, 2018; Tomczyk & Borowska-Stugińska, 2018; Tomczyk, 2018).

We used skeletal material from Radom as it contains all three types of morphological variations: PO, AL, and FP. In research on morphological differences, it is important to analyse individuals from the same population to avoid the possible influence of biological or environmental factors (McCoy et al., 2006; Wolański, 2008). Moreover, the studied variants of the femoral head-neck junction can be related to a variety of habitual behaviours (Molleson, 2007). That is why the analysis of a homogenous population ensures that possible differences in morphology will not be biased. The population from Radom was selected for the research due to the relatively large sample size (346 individuals) and fairly well-preserved hip joints, as for skeletal material. Figure S1 presents a flowchart with sample selection information.

Standard anthropological methods by Buikstra & Ubelaker (1994) were used to determine the sexes of skeletons. In most cases, this included visual assessments of pelvic and cranial features. The age-at-death estimate was based on changes in the morphology of the pubic symphysis, using the Brooks & Suchey (1990) system and standards for changes in the topography of the auricular surface (Buikstra & Ubelaker, 1994).

Only adult remains were included in this study. To avoid an influence on PO, AL and FP formation and its interpretation, only individuals without any other observable skeletal changes (such as observable bone illnesses, traumas, fractures or bone deformities) were included. Considering all the above limitations (complete femoral and pelvic bones, especially in the acetabulum area, without any visible changes), 52 sets of bones (femur and pelvic bone) were included in the final analysis (seven sets of bones without any head-neck junction changes, 15 with noted PO, 16 with FP and 14 with AL). Due to poor preservation of the skeletal material and low frequency of PO, FP and AL, analyses were performed without sides and sex division. Usually, right bones were included. If right bones were not available or damaged bones of the left skeletal side were used. In several individuals, the femur and acetabulum were preserved on both sides, but to avoid artificially decreasing the sample variation, only one side (the right) of these individuals was included in further analysis.

All three types of the head-neck joints were noted according to recommendations by Radi (2014). In a study by Radi (2014), the definition and prescription of the cribra femoris recording method coincides with the definition of AL by some other authors (see Göhring, 2021). Therefore, in our study, AL is analysed according to Göhring (2021) recommendation. Because cribra femoris is not a nonmetric trait but a pathological condition (Göhring, 2021) this trait was not taken into account in the present analyses. Taking the above it is important to underline here that in order to avoid errors in classification of the head-neck junction features and thus improper results and its interpretation, investigators in this field should be vetted. It is an essential key to the correct and reliable comparison of morphological results of head-neck junction changes in different studies. The assessment of head-neck junctions was carried out by one observer (AM).

Computed-tomography scanning

Fifty-two femur-pelvis sets were scanned using a 32-slice computed tomography (Siemens SOMATOM Sensation) in the cranio-caudal position and a standard protocol (0.625 slice thickness and 120 kV). The same approach was used in previous research (e.g., Kubicka, Myszka & Piontek, 2015; Kubicka et al., 2022). Next, three-dimensional (3D) models of the femur and the pelvis were performed from each CT scan using 3D Slicer software (version 5.0) (Fedorov et al., 2012) with the Threshold tool using similar segment values within the sample (i.e., 52 sets of bones).

Linear and angle measurements

The measurements described in Table 1 and shown in Fig. 1 were obtained twice by the one observer (AMK) on 3D bone models using Gom Inspect software (version 2.0.1). These parameters are commonly used to describe the overall morphology of the femur in orthopaedic research (e.g., Moats et al., 2015; Musielak et al., 2021; Rogers et al., 2021). In turn, pelvic measurements focused on the acetabulum due to insufficient bone preservation. The parameters were measured in millimetres with decimal precision.

Table 1 Description of the measurements.

Measurement	Abbreviation	Bone	Description	
Height of the head	SI head	Femur	The distance between the most superior and inferior point on the femoral head	
Height of the neck	SI neck	The distance between the most superior and inferior point on the femoral neck	
Proximal breadth		The distance between the most medial point on the head and the most lateral	
Apparent neck-shaft angle	aNSA	The angle between the femoral neck axis and the long axis of the femoral shaft, measured on the apparent anteroposterior view	
True neck-shaft angle	tNSA	The angle between the femoral neck axis and the long axis of the femoral shaft, measured on the true anteroposterior view	
Inclination angle	Incl. angle	The angle between the femoral head and a plane located on the greater trochanter and the femoral condyles, measured on the version view	
Alpha angle	Alpha	The angle between the head surface and the femoral neck, measured on the version view	
Version angle	Ver. angle	The angle between the femoral neck axis and a line parallel to the posterior aspect of the femoral condyles, measured on the version view	
Acetabulum area (mm2)	Area	Pelvis	The area of the acetabulum	
Acetabulum depth		The distance between the most inferior point of the acetabulum to the point on the surface that lies on the acetabulum border	
Acetabulum height	SI acetabulum	The height of the acetabulum measured from the most superior to the most inferior point	
Acetabulum width	AP acetabulum	The width of the acetabulum from the most anterior to the most posterior point	

Figure 1 Measurements of the femoral head and the acetabulum.

(A) Height of the femoral head; (B) height of the femoral neck; (C) proximal breath, (D) apparent neck-shaft angle; (E) true neck-shaft angle; (F) version angle measured between points a, b and c; (G) alpha angle measured between points a, b and c; (G) inclination angle measured between points b, c and d; (H) acetabulum area; (I) acetabulum depth; (J) acetabulum height; (K) acetabulum width.

Geometric morphometrics

Shape analysis of the femur and pelvis was also performed using the geometric morphometric approach. Fifteen landmarks and 25 surface semilandmarks were digitised on the proximal end of the femur. In turn, one landmark and 63 semilandmarks were distributed on the acetabulum area. Semilandmarks were evenly distributed along the edge of the articular surface of the femoral head and acetabulum using 3D Slicer software. This part of the study was carried out by one observer (AMK). A detailed description and localisation of the landmarks and semilandmarks are shown in Fig. 2 and Table 2.

Figure 2 Localization of landmarks (red points) and semilandmarks (blue points) digitised on the bones.

(A) Anterior view of the femur; (B) medio-posterior view of the femur; (C) lateral view of the femur; (D) lateral view of the pelvis.

Statistical analysis

The accuracy of linear and angle measurements was assessed using the intra-class correlation (intra-ICC) coefficients, whose values close to 1 indicate excellent reliability and values close to 0 indicate no reliability of measurements. Next, the measurements were tested for normality and homogeneity of variance using the Shapiro–Wilk test and Bartlett’s test, respectively. All parameters are not significantly different from normal distribution and show homogeneity of variance; therefore, a further statistical analysis was performed using the parametric test. ANOVA with a post-hoc test (i.e., Tukey’s HSD) was performed to analyse the differences in measurements of the femur and acetabulum between bones with femoral-neck changes and bones without them.

In the case of geometric morphometric analysis, Generalised Procrustes Analysis (GPA) was performed on the raw coordinates to superimpose landmarks and semilandmarks (Zelditch et al., 2004). Next, a multivariate regression with a permutation test of 10,000 rounds between the Procrustes coordinates (shape variables) and the centroid size (size variable) was calculated to check for allometry. The association between the shape and size was significant in the femoral head (p < 0.05, RV = 30.34%) but not in the acetabulum (p > 0.05, RV = 3.34%). A major part of the variation in the femoral shape was influenced by the bone size; therefore, further geometric morphometric analysis was performed on the regression residuals of the femur. This approach allowed us to exclude the influence of size on the shape. Next, principal component analysis (PCA) was performed to quantify the variation in the shape of the femoral head and the acetabulum. In turn, canonical variate analysis (CVA) with a permutation test of 10,000 rounds was performed to analyse differences in bone shape between three types of pathologies and healthy hip joints. Statistical analysis was performed using R software and Evan Toolbox software (version 1.75). The raw data containing raw coordinates and measurements can be found in the Mendeley Repository under the DOI: 10.17632/5kswnkfdsd.1.

Results

Linear and angle measurements

The final statistical analysis focuses on 52 sets of bones (femur and pelvic bone). The intra-ICC coefficients of all measurements were higher than 0.98. Descriptive statistics are shown in Table 3. ANOVA exhibited significant differences between the three types of analysed changes (i.e., AL, FP and PO) and the neck-head junction without any changes, only for the height of the femoral head (p < 0.036). Further, Tukey’s post-hoc test exhibited significant differences in the height of the femoral head only between the morphological variations (Table 4).

Geometric morphometrics

PCA showed that femurs of hip joints with and without changes showed similar variation in the shape of the femoral head (Fig. 3A). PC1 and PC2 explained 44.73% and 23.00% of the shape variation, respectively. Femurs with positive values of PC1 were characterised by a greater trochanter located closer to the femoral head. In turn, femurs with positive values of PC2 exhibited a more elevated superior part of the greater trochanter. The CVA showed no significant differences in the shape of the femoral head between types of the head-neck joints (Fig. 4A, Table 5, all p > 0.05). CV1 and CV2 are responsible for 54.78% and 28.86% of the shape variation in the femoral head.

Table 2 Description of landmarks and semilandmarks.

Point	Type	Bone	Description	
1	LM	Femur	Point at the fovea of the femoral head	
2	Deepest point of trochanteric fossa	
3	Most superior-anterior point at the greater trochanter	
4	Most posterior point at the greater trochanter	
5	Most lateral point at the greater trochanter	
6	Most anterior point at the lesser trochanter	
7	Most posterior point at the greater trochanter	
8	Most inferior-anterior point at the greater trochanter	
9	Most superior point at the femoral neck (i.e., midshaft)	
10	Most superior point at the lesser trochanter	
11	Most inferior point at the lesser trochanter	
12	Most inferior point at the femoral neck	
13	Most posterior point at the lesser trochanter	
14	Most inferior-posterior point at the greater trochanter	
15	Most superior point at the femoral head	
1-25	sLM	Semilandmarks around the femoral head (along the edge of the auricular surface)	
1	LM	Pelvis	Deepest point of the acetabulum	
1-63	sLM	Semilandmarks around the acetabulum (along the edge of the auricular surface including the lunate surface)	
Notes.

LM landmark

sLM semilandmarks

Table 3 Descriptive statistics of measurements performed on the femur and pelvic bone.

Head-neck changes		Femur	Pelvis	
		SI head	SI neck	Proximal breadth	aNSA	tNSA	Inc. angle	Alpha	Ver. angle	Area	Acetabulum depth	SI acetabulum	AP acetabulum	
Allen’s fossa	Mean	44.88	34.93	92.87	132.29	129.85	10.34	55.98	10.15	3,963.76	24.94	60.04	56.14	
Sd	3.73	4.33	7.73	5.17	5.53	5.15	3.96	4.96	618.15	2.64	4.91	4.55	
Femoral plaque	Mean	48.67	37.64	99.02	130.64	129.96	12.49	59.96	13.62	4,701.55	25.91	63.68	59.98	
Sd	3.61	2.84	6.67	5.52	5.68	4.31	7.26	4.11	771.93	3.49	3.56	3.92	
Porrier’s facet	Mean	47.92	35.83	96.92	130.45	125.77	14.93	57.61	15.13	4,482.94	25.90	62.76	59.53	
Sd	3.72	3.37	8.01	6.02	6.11	5.76	5.00	6.31	710.49	3.55	4.40	3.75	
No changes	Mean	46.27	33.67	93.04	128.70	127.04	12.69	59.77	12.89	4,348.13	25.68	61.81	58.53	
Sd	3.98	4.28	6.96	6.55	7.55	7.27	6.67	8.46	550.19	1.91	3.77	4.18	
Notes.

Sd standard deviation

SI head height of the head

SI neck height of the neck

aNSA apparent neck-shaft angle

tNSA true neck-shaft angle

Inc. angle inclination angle

Alpha alpha angle

Ver. Angle version angle

Area acetabulum area (mm2)

SI acetabulum acetabulum height

AP acetabulum acetabulum width

Table 4 Post-hoc tests, Tukey HSD, for height of the femoral head.

Trait	Mean difference	p-value	Lower bound	Upper bound	
Allen’s fossa	Femoral plaque	3.78	0.047	0.03	7.53	
	No changes	1.39	0.851	−3.20	5.98	
	Porrier’s facet	3.04	0.123	−0.54	6.62	
Femoral plaque	No changes	−2.39	0.514	−6.99	2.20	
	Porrier’s facet	−0.75	0.945	−4.33	2.83	
Porrier’s facet	No changes	1.65	0.759	−2.81	6.10	
Notes.

Bold indicates significant differences at the level p < 0.05.

The PCA of the acetabulum of hip joints without changes is characterised by lower variation than the acetabulum of hip joints with AL, PO or FP (Fig. 3B). PC1 and PC2 are responsible for 50.50% and 15.65% of the shape variation in the acetabulum. Positive values of PC1 show a more indented anteroposterior part of the lunate surface, while positive values of PC2 are responsible for a more indented inferior edge along its entire length (i.e., from the anterior to the posterior part). In terms of the types of head-neck joints, CVA showed no significant differences in shape of the acetabulum (Fig. 4B, Table 5, all p > 0.05). CV1 and CV2 explained 54.57% and 28.75% of the shape variation in the acetabulum.

Discussion

The results do not show significant differences in morphology between joints with AL, FP and PO and hip joints without these bony changes. Joints with and without observable changes in the head-neck junction region show similar variations in the shape of the femoral head. The only difference is seen in the height of the femoral head between the FP and AL (Table 4).

In skeletal material from Radom, the major part of shape variation in the femur (i.e., PC1) is explained by the location of the greater trochanter. Assuming that excessive adduction and abduction in the hip joint is the cause of changes in the femoral head and neck (Mei-chao et al., 2005; Ollivier et al., 2017), the action of the muscles attaching to the greater trochanter (e.g., the gluteus medius) influenced the shape and size of the trochanter and its formation. If it is located closer to the head of the femur, the greater trochanter shows a more raised upper part. Several authors show the existence of a direct relationship between these skeletal variants of the proximal femur and habits associated with exercise or rest (see Villotte & Knüsel, 2009; Wagner et al., 2012; Radi et al., 2013; Mellado & Radi, 2015; Bühler & Kirchengast, 2022).

Figure 3 PCA of the variation in the shape of the femur (A) and acetabulum (B).

Figure 4 CVA of the shape of the femur (A) and acetabulum (B) between the hip joints with changes (i.e., Allen’s fossa, Poirier’s facet, femoral plaque) and without changes.

In our sample, the acetabulum of the hip joint without observed changes was characterised by a slightly lower variation than the acetabula in the hip joints with PO, AL or FP. Hip joints with PO, AL or FP show a more indented anteroposterior part of the lunate surface and an indented inferior edge along its entire length (i.e., from the anterior to the posterior part). It could be hypothesised that the acetabulum shape may be a result of the appearance of head and neck changes that are an effect of excessive flexion/extension movement during various activities (horse riding, intense hip adduction), a bony reaction to bony pressure.

Radi (2014) underlines that the variable appearance of head-neck junction changes could be related to individual variability in reacting to pressure as well as in the dimensions of the proximal femur (neck and head dimensions, neck-shaft angles) and of the overlying soft tissues. Unfortunately, the available archaeological, historical and biological data do not allow us to assess the model of physical activity of the Radom population (see Material and Methods section). It might be worth performing similar analyses on a well-defined population in terms of physical activity.

The unknown aetiology, lack of a clear definition and terminology for femoral head-neck junction changes, similar localisation and overlapping areas of bone lesions can be reasons for the lack of significant differences in a bone morphology between joints with AL, FP, PO and hip joints without these changes in the present study. Moreover, these three main variations of femoral head-neck junction formation do not cover the whole spectrum of possible variations or represent a validated classification (Mellado & Radi, 2015) which additionally affects the difficulties in the evaluation and analysis of data, making it impossible to compare them and, consequently, to interpret them and formulate final conclusions.

It should be mentioned here that the present study results could have been affected by the small sample size. Due to damage, incompleteness or missing bones, only a small part of the skeletal material from Radom could be taken into the analyses (only skeletons with complete set of bones—femur and pelvic bone on the same side). However, it should be mentioned that analysis of individuals from the same population assures us that the results obtained are the actual effect of morphology resulting from the type of hip joint and not due to environmental and genetic differences between individuals from various populations. On the other hand, populations can differ from each other in the degree of morphological variation. Therefore, it is possible that we did not detect the differences between the types of hip-joints due to small morphological variation in our analysed population. We suggest that similar studies should be made on other and more numerous skeletal material.

Table 5 CVA with the permutation tests between hip joints with changes (i.e. Allen’s fossa, Poirier’s facet or femoral plaque) and without changes.

Trait	Femur	Trait	Pelvis	
	AL	FP	NO		AL	FP	NO	
FP	0.057			FP	0.353			
NO	0.550	0.620		NO	0.784	0.974		
PO	0.080	0.646	0.840	PO	0.148	0.445	0.663	
Notes.

AL Allen’s fossa

FP Femoral plaque

NO hip joint without changes

PO Poirier’s facet

Conclusions

In summary, geometric morphometrics and measurements showed similarities in the shape of the hip joints with and without changes in the head-neck junction region. Therefore, the femoral head-neck junction changes do not significantly affect the joint anatomy (no differences between joints with noticed Poirier’s facet, Allen’s fossa or femoral plaque and without changes). A smaller variation in the acetabulum shape of the head-neck junction without changes than in joints with changes can indicate that this region is under pressure from the femur. However, an understanding of the role and efficiency of that influence needs further studies.

Supplemental Information

Supplemental Information 1 List of a DOIs numbers

Supplemental Information 2 Flowchart showing the selection of bones for analysis

We thank Darek Kurkiewicz from the Clinical Hospital in Oborniki for technical support in scanning the osteological material.

Additional Information and Declarations

Competing Interests

Author Contributions

Human Ethics

Data Availability

The authors declare there are no competing interests.

Anna Myszka conceived and designed the experiments, performed the experiments, authored or reviewed drafts of the article, receive funding, and approved the final draft.

Anna Maria Kubicka conceived and designed the experiments, performed the experiments, analyzed the data, prepared figures and/or tables, authored or reviewed drafts of the article, and approved the final draft.

The following information was supplied relating to ethical approvals (i.e., approving body and any reference numbers):

No administrative and ethical permissions were obtained because (1) according to Polish law there is no need to obtain permission from a biological commission when research is carried out on historical osteological collections, (2) no new archaeological excavations were carried out to gain the osteological material, (3) the research protocol used in this study was already tested by other researchers. The authors confirm that all stages of the research (collection, CT scanning and analysis) were performed in accordance with the regulations on the analysis of human remains.

The following information was supplied regarding data availability:

The coordinates of LM distributed on the femur and pelvis, and the measurements of all analysed bones is available at Mendeley: Kubicka, Anna Maria (2025), “Femoral head-neck junction changes in skeletal material”, Mendeley Data, V1, doi: 10.17632/5kswnkfdsd.1.

The 3D models are available at Morphosource: https://www.morphosource.org/projects/000740996?locale=en.

The 95 DOIs for each specimen are available in the Supplemental File.

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
