# Peer review of "Morphological variations of the femoral head-neck junction in historical skeletal material"

_PeerJ, doi:10.7717/peerj.20236_

## Round 0.1 · original submission · Major Revisions

Two reviewers agree that the manuscript requires Major Revisions, however both have provided detailed comments and suggestions on how to improve the manuscript. I look forward to receiving your revised submission once the Reviewers' concerns have been addressed.

Reviewer 1 ·

Basic reporting

Title & Abstract

Title: I recommend that the authors provide the research design in the title to improve the clarification of interpretation. For example: “Morphological variations of the femoral head-neck junction: a historical anatomical analysis” or something to that effect.

Abstract:
Background: Page 6, Line 18: I recommend the authors consider changing the word ‘dependency’ to ‘association’ as this would provide a wider evaluation of this research question.
Methods: Page 6, Line 23: I recommend the authors provide some information on their principal analysis, i.e., what the principal research question analyzed was and what statistical analyses were undertaken.
Results: Page 6, Lines 24-32: The authors should provide data to support the summary findings. For example, the authors need to quantify what was significant, i.e., p-values and descriptive data. I recommend they provide descriptive data to quantify ‘increased’ and ‘more’, and ‘changes’ in this section. This would provide more detail and accuracy in the reporting in this study.
Results: Page 6, Line 32-35: This information is more ‘Conclusions’ and therefore I recommend this be included in an Abstract, Conclusions section rather than in the Results. This would provide greater clarity.

Introduction
The Introduction section is clear and well presented. It is supported by appropriate, contemporary evidence. The argument for the research question is clearly communicated. My only reservation is regarding the final paragraph. I think the authors can summarize the rationale for the study, present the aims of the paper, and then conclude this section. The information provided in the following points is aimed at providing a more concise final paragraph.
Page 8, Line 83-86: These sentences provide more explanation of the findings and not a context or introduction to the research question. For clarity, I recommend that these be removed.
Page 8, Line 89-96: Equally, the information provided in these sentences is Methods and is going to be repeated in the following section. For clarity of message, I recommend that they be removed, as these are redundant.

Figures & Tables
The figures and tables are all clear and legible. There are no unnecessary modifications to the images. I have nothing further to recommend.

Experimental design

Materials and Methods

Page 8, Line 102: Because of the ‘sensitive’ nature of this data collection approach, I recommend the authors insert the sentence on their ethical approvals/study approvals in this sentence to clarify that all research sample collection was approved prior to conducting. Through this, I recommend that the authors move the information from Lines 136 to 141 to this section.
Page 9, Line 104-106: I recommend the authors provide a citation for the statement “In research on morphological differences, it is important to analyze individuals from the same population to avoid the possible influence of biological or environmental factors”- this will provide the transparency of this important statement on sampling approach.
Page 9, Line 107-108: I recommend the authors reconsider the statement “That is why the analysis of a homogenous population ensures that possible differences in morphology will not be biased” – please explain how this ensures an unbiased population. This is crucial for justifying the sampling strategy. The authors should explain on if they are referring to epigenetic variation or behavioral impact, or what to underscore this point.
Page 9, Line 122-125: The authors state “Due to poor preservation of the skeletal material and low frequency of Poirierís facet, femoral plaque and Allenís fossa, analyses were performed without sides and sex division” And then explain “To avoid lower variability of the above traits manifestation, only one (right or left) set of bones (femur and pelvic bone) in each individual…” – please clarify in the text what this means. It is unclear if the authors mean as it later becomes clear that they only included one side for the 52 individuals. I recommend that the authors review these two sentences and ensure that the meaning is clearer for the reader to understand. A flow chart of how the 52 samples were gathered from the overall cohort may be helpful to communicate this information. I recommend that the authors consider this addition as well.
Page 10, Lines 143-163: The authors clearly report the different imaging and non-imaging measurements. These are all appropriate for the research question. However, I recommend that the author explicitly state who performed these assessments, whether they were completed by multiple different assessors, and the reliability of these measurements. This would improve the robustness of assessment rigor.

Statistical Analysis, Page 11, Lines 166-185: The statistical analysis approach was robust and appropriate to answer the research question. All statistical assumptions to justify the selection of these analysis approaches were met. I have no further comments on this section.

Statistical Analysis, Page 11, Line 185: A link to the raw data repository is presented in the paper. This is a valuable addition.

Validity of the findings

Results

The authors have appropriately reported their findings. However, there is a need to provide greater depth in the results text to fully appreciate the findings. These are highlighted in the following points:
Page 11, Line 187: I recommend that the authors start the Results section with an overview of the samples and characteristics of their 52 specimens. This would provide a clear picture of the cohort presented.
Page 11, Line 189: The authors state, “ANOVA showed significant differences between the three types of analyzed changes” – I recommend they provide descriptive and p-value data to support these statements.
Page 12, Line 198-199: The author states, “In turn, femurs with positive values of PC2 exhibited a more elevated superior part of the greater trochanter” – I recommend the authors provide data to support what ‘elevated superior’ is. I recommend the addition of descriptive data to quantify the magnitude of this elevation.
Page 12, Line 204-206: The authors state, “The CVA showed no significant differences in the shape of the femoral head and hip joint between types of hip joint morphology” – to improve interpretation, I recommend the authors provide descriptive statistics and p-values to support these reported findings.
Table 3: I recommend that the authors consider whether the data can be presented to two and not three decimal places. This would clarify reporting.

Discussion

Page 12, Line 209-233: The summary of findings presented in the opening Discussion paragraph is clear and an appropriate interpretation of the findings.
Page 12, Line 215-216: The authors state “Assuming that excessive adduction and abduction in the hip joint is the cause of changes in the femoral head and neck” – this is a plausible explanation for the results but for clarity I recommend eh authors provide a citation/reference to support the assumption made. This will strengthen their argument.
Page 13, Line 235-241: The author in this paragraph explores the level of sphericity of the femoral head. However, there is no commentary or interpretation of these parameters in the results from this analysis. To ensure that this point is comprehensively considered in relation to these new findings, I recommend a direct comparison, and interpretation is provided in this paragraph; otherwise, the value is less obvious to the reader.
Page 13, Line 242-249: I am unsure how the argument of bipedal locomotion applies in this Discussion. The samples analyzed are not for quadrupedal specimens. All specimens presented will have been bipedal. The consideration of physical activity is valid, although challenging to understand given the lack of data presented, but the argument for bipedal and variance thereof is less relevant. I recommend that the authors consider the message of this paragraph and either refine this or reconsider its value.
Page 13, Line 258: The authors should consider the issue of external validity and whether the fact that all specimens arose from one location/population impacts the generalizability of the study, and is an important limitation. For completeness, I recommend the authors consider this in their study’s limitations section.
Page 13, Line 258: The issue of lack of corresponding data on physical activity levels, diet, and genetics may also be a further limitation to consider. I recommend the authors consider this in their study’s limitations section.

Conclusion

Page 14, Line 264-270: The authors provide a clear and concise Conclusion section. The interpretation is appropriate. The caveat based on limitations is also appropriate. I have nothing further to recommend to this section.

·

Basic reporting

The manuscript presents a morphological analysis of the proximal femur and acetabulum, involving canonical skeletal features such as Poirier’s facet, Allen’s fossa, and femoral plaque. Some comments:

Line 47: The paragraphs provide detailed descriptions of anatomical structures, including Poirier’s facet, Allen’s fossa, and femoral plaque; these should be accompanied by a corresponding figure. Moreover, the manuscript should clarify how these features have the potential to relate to the variables being analyzed (e.g., shape and size of the acetabulum and proximal femur, Line 87).

Line 421: If abbreviations are permitted, they should be used consistently throughout the manuscript to avoid redundancy and improve readability.

Experimental design

Line 87: The authors state that samples come from the "same population" based on controlled habitual, biological, and environmental factors. It should be clarified whether samples from Radom can scientifically be regarded as a homogeneous population, and if possible, what criteria were used to confirm this.

Line 116: Even in cadaveric studies, a flowchart or diagram outlining the sample selection process is essential to ensure transparency and to mitigate potential selection bias. The absence of a power analysis, combined with the limited sample size and lack of statistical significance, raises concerns. It should also be clarified whether the samples were collected consecutively and comprehensively. Specifically, please confirm whether the 7 bones without head-neck changes represented all eligible samples within that category.

Line 147: The segmentation methodology should be described in more detail. Was segmentation performed solely using the threshold tool of the same segment value in the Segment Editor? If not, this introduces a risk of subjective bias. Please disclose the experience level of the annotator or whether any quality control or validation process was implemented.

Line 150: As these measurements were derived from region-based anatomical landmarks with potentially subjective definitions, an inter- or intra-rater variability analysis is necessary. If such analyses were not conducted, please provide a justification.

Validity of the findings

In Figure 4A, the visual differences were observed, such as the green-marked femoral plaque samples. However, the manuscript does not offer any follow-up analysis or discussion regarding these observations.

Although some comparisons did not reach statistical significance, possibly due to the limited sample size, statistical non-significance does not imply a lack of meaningful trends. For instance, comparisons between the Allen’s fossa and Poirier’s facet groups (Table 3) warrant further investigation with an expanded sample size.

The discussion highlights some biomechanical considerations, such as the influence of motion (e.g., adduction and abduction) on femoral morphology (Line 215). These are interesting and could serve as a foundation for future biomechanically informed investigations.

---

## Round 0.2 · accepted · Accept

Thank you much for seriously addressing the reviewers' concerns and providing a detailed response. We sent your revisions to both authors who suggested Major Revisions, and they agree the paper is ready to be accepted. The first reviewer requested you reconsider changing the title, but from your previous response I see you prefer not to. You can take the reviewer's comment under advisement, but I don't see this as a determining factor. You paper is accepted. Congratulations on your nice work.

Reviewer 1 ·

Basic reporting

Title & Abstract
Title: I recommend again that the authors provide the research design in the title to improve the clarification of interpretation. For example: “Morphological variations of the femoral head-neck junction: a historical anatomical analysis” or something to that effect. The current title is ambiguous for people reviewing this paper and therefore may reduce the impact of this paper. I recommend the authors reconsider this.
Abstract: Page 8, Lines17-38: The Abstract is clear and informative. There is sufficient information presented to both understand the research question, clearly appreciating the study design and associated results, and interpret the Conclusions. The addition of data to the Results section now allows interpretation of the Conclusions with greater assurance. I have nothing further to recommend to this section.

Introduction
Introduction, Page 9, Lines 42 to Page 10 Line 99: The Introduction provides a clear overview of the clinical and anatomical areas. This is underpinned with relevant evidence. The interpretation offered by the authors is appropriate and robust. The rationale for the study is clear and research question and purpose appropriately stated in Lines 96 to 99. I therefore have nothing further to recommend to this section.

Figures & Tables
The figures and tables are appropriate, clear and legible. The figures are free from unnecessary modifications.

Experimental design

Material and Methods
Materials and Methods, Page 10, Lines 101 to Page 13, Line 201: The Methods applied are appropriate to answer the research question. The data source and measures used are clearly communicated and appropriately justified. The statistical analyses adopted are also appropriate. I have nothing further to recommend to this section.

Validity of the findings

Results
Results, Page 14, Line 204 to Page 226: Whilst the Results are briefly reported in this section, this is appropriate, as the authors have now provided greater data both in the results section but also across the different figures and tables. These are all relevant and accurately interpreted against the data and research question. There is transparency in the presentation. I have nothing further to recommend.

Discussion
Discussion, Page 14, Lines 229 to Page 16 Lines 272: The Discussion is clear and focused. This is underpinned with relevant and credible associated citations and literature. The interpretation reflects the data presented in the Results section and is not over-interpreted. The limitations point on same cohort as being a strength is acceptable as the rebuttal from the authors in their response document is understandable. I think this is appropriately communicated in the Discussion section too (Lines 266 to 270) with criticality in this point. I therefore have nothing further to add to this section.

Conclusion
Conclusion, Page 17, Line 275 to 281: The Conclusion section is clear, focused and appropriately interpreted. The interpretation does not overstate the findings and there is clear translation to both the research question and study analyses. I have nothing further to recommend on this as it currently stands.

Additional comments

None

·

Basic reporting

Ok.

Experimental design

Ok.

Validity of the findings

Ok.

Additional comments

Apart from the incorrect line numbering in the response letter, I am satisfied with the authors’ response.